# Enantioselective de novo construction of 3-oxindoles via organocatalyzed formal [3 + 2] annulation from simple arylamines

Yong Wang [1], Yanyan Li[1], Haohua Chen [2,3] ✉, Yu Lan [1,3], Chao Pi[1,3], Yangjie Wu[1,3] & Xiuling Cui [1,3] ✉

The de novo construction of enantioenriched 3-hydroxyindolenines and 3-oxindoles from easily available starting materials has been highly desired. Herein, an enantioselectively intermolecular direct [3 + 2] annulation of aryl amine with 2,3-diketoesters to construct 3-hydroxyindolenines with a chiral tertiary alcohol has been disclosed. The results of control experiments and DFT calculation revealed that π − π interaction plays a pivotal role in the enantioselectivity-determining process of [3 + 2] annulation. The following unusual concerted [1,2]-ester migration provides a family of chiral 3-oxindoles in good to excellent yields with excellent enantioselectivity.

Chiral 3-oxindole is a prominent structural motif found in numerous natural products[1–4] and exhibits extensive bioactivities[5–8] and fluorescence property[9]. Continuous endeavors have been devoted to the enantioselective synthesis of such moiety starting from indole and its derivatives during the past two decades. For instance, Kawasaki, Xu, Hou and Chan groups developed the protocols for chiral 3-oxindoles through the reaction of 2-monosubstituted 3-oxindoles with a series of electrophiles in the presence of cinchona analogs or palladium/chiral phosphine ligand as the chiral auxiliary (Fig. 1a)[10–13]. You group successfully built chiral 3-oxindoles starting from racemic spiro indolin-3-ones as the electrophiles through chiral phosphoric acid (CPA) catalyzed enantioselective Friedel–Crafts alkylation of indoles (Fig. 1b)[14]. Glorius and Xu's groups independently described the synthesis of chiral spiro-3-oindoles via organocatalyzed formal [3 + 2] annulation of aurones with 1,3-dipoles (Fig. 1c)[15,16]. With 2-monosubstituted indoles as the starting materials, considerable achievements on the catalyzed enantioselective 3-oxindole synthesis have been contributed via the indole oxidation and enantioselective nucleophilic addition process under organo-catalysis (Fig. 1d)[17–24]. Xiao and Jiang's groups developed a CPA catalyzed enantioselective 3-oxindole synthesis via the photo-induced aerobic oxidation and semi-pinacol rearrangement independently (Fig. 1e)[25,26]. In spite of the above elegant achievement, de novo simultaneously building 3-oxindole ring and a chiral quaternary carbon via asymmetric catalysis has been far behind.

Herein, we disclosed a de novo enantioselective synthesis of 3-oxindole starting from simple arylamines and 2,3-diketoesters[9]. A formal [3 + 2] annulation catalyzed by CPA followed by rare [1,2]-ester migration was involved in this transformation (Fig. 1f)[10]. To realize this de novo construction of chiral 3-oxindoles, main problems as follows should be overcome: (1) the low reactivity of ketone[27–29]; (2) the ortho-selectivity of C-H addition should be controlled since para-selectivity is usually preferred[30–34] with para- non-hindered aryl amines[35–38]; (3) difficulty in 1,2-ester migration; (4) the enantioselective control of [3 + 2] annulation and 1,2-ester migration. The DFT calculation revealed that the CPA catalyzed ortho-C-H addition with ketones[39] was the rate- and enantioselectivity-determining step for the [3 + 2] annulation process. And an exclusive enantioselective [1,2]-ester migration[40,41] occurred through a concerted three-membered ring transition state.

## Results and Discussion: Investigation of reaction conditions

1-Naphthylamine **1a** and 2,3-diketoester **2a** were chosen as the substrates to initiate our investigation with (*R*)-**5a** as the catalyst, toluene as solvent, at room temperature for 40 h. The **3aa** was isolated and heated for another 20 h and (*S*)-**4aa** was obtained in 95% yield with 45% ee. A family of chiral BINOL CPAs as the catalysts were then examined (Table 1, entries 1-4). The bulky BINOL-(*R*)-CPAs and condensed aromatics substituted BINOL-(*R*)-CPAs produced opposite enantiomer **4aa**, which indicated that different coordination patterns might exist

[1]Henan Key Laboratory of Chemical Biology and Organic Chemistry, College of Chemistry, Zhengzhou University, Zhengzhou, Henan, PR China. [2]State Key Laboratory of Antiviral Drugs, Henan Normal University, Xinxiang, Henan, PR China. [3]Pingyuan Laboratory, Henan, PR China. ✉e-mail: chenhaohua@htu.edu.cn; cuixl@zzu.edu.cn

**Fig. 1 | Construction of chiral 3-oxindoles. a–e** Previous synthesis of chiral 3-oxindoles from indole derivatives. **f** This work: enantioselective de novo construction of 3-oxindoles from arylamines. CPA chiral phosphoric acid.

for the bulky and condensed aromatics substituted CPAs in this catalytic system. Firstly, the sterically bulky CPAs were surveyed (Table 1, entries 5-7). The bulkier BINOL-(*R*)-CPAs gave the product **4aa** with better enantioselectivity (Table 1, entries1-2, 5-7). Especially, (*R*)-**5c** with 2,4,6-trisisopropylphenyl group at the *para*-position of the aryl ring in BINOL-(*R*)-CPA produced (*S*)-**4aa** with 81% ee (entry 7). To accelerate the reaction rate at room temperature of step b, TsOH·H$_2$O was added as promotor (entries 5-9). The evaluation of different solvents showed that the chlorinated hydrocarbon gave better enantioselectivity with the bulky CPAs (Supplementary Table 1 in Supplementary Information). The combination of CH$_2$Cl$_2$ and ODCB (1,2-dichlorobenzene) (4:1) improved the ee to 88% (entry 8). Increasing the amount of solvent (CH$_2$Cl$_2$: ODCB = 4:1) to 4 mL, 91% ee of (*S*)-**4aa** was obtained (entry 9). Thereafter, the further evaluation of spirocyclic-based CPAs (entries 10-12) showed that 9-anthryl spirocyclic CPA (*R*)-**6b** gave good enantioselectivity (68% ee). Solvents were surveyed with (*S*)-**6b** as the catalyst (entries 13-14) and AcO$^i$Pr gave the best enantioselectivity (88% ee). Increasing the amount of AcO$^i$Pr to 4 mL, 92% ee of (*S*)-**4aa** was obtained (entry 15).

## Scope

With the optimized reaction conditions in hand, the substrate scope was next examined (Fig. 2). 2,3-Diketoesters **2** with alkyl (R) groups, including methyl, *n*-butyl, benzyl and *n*-heptyl, reacted smoothly with **1a** to generate **4ab-4ae** with high yields and enantioselectivity. The *para*-substituted halogen (**2f-2h**), electron-withdrawing groups (**2i-2k**) and methyl (**2 l**) in phenyl group of **2** gave the desired products with excellent yields (88–99%) and enantioselectivity (94-96%). -OMe group (**2 m**) resulted in **4am** with 78% ee and 97% yield using 0.5 mL of solvent. The substrates **2** with *meta*-substituted group could afford the product with excellent yield and ee (**4an**, 90% yield, 94% ee). The *ortho*-substituents gave product with good yield and low ee (**4ao**, 80% yield, 20% ee). Gratifyingly, using (*R*)-**5c** as the catalyst, 4:1 ratio of CH$_2$Cl$_2$ and ODCB as the solvent, (*S*)−**4ao** was obtained in excellent yield with 86% ee. Substrates **2** with 2-furyl or 2-thienyl groups produced **4ap** and **4aq** with excellent ee (96% and 97%). A successful gram-scale synthesis

of **4aq** demonstrated the practicality of this reaction. The alkenyl substituted diketoester was applicable for this transformation (**4ar**, 85% yield, 90% ee). Then, the tolerance of substituents on the 1-naphthylamine were evaluated. Cyclopropyl (**1b**), phenyl (**1c**), 1-naphthyl (**1d**), 3-thienyl (**1e**), bromo (**1f, 1g**), chloro (**1h**) and methoxy (**1i**) groups at the different position were tolerated well and the corresponding products were provided smoothly with good to excellent yields (61-99%) and excellent enantioselectivity (94-99% ee) using **2q** as the reaction partner. Among them, product **4dq** with two stereogenic centers of a chiral quaternary carbon and an axially chiral center (1:1 dr). In addition, 1-aminoanthracene (**1j**) worked well, providing **4jq** in 93% yield and 97% ee. Interestingly, the reaction of 3-fluoranthenamine (**1k**) and **2q** produced the opposite enantiomer product **4kq** in 99% yield and 96% ee. 1-Dibenzofuranamine (**1l**) also produced the desired product **4lq** with excellent ee of 96% in moderate yield. Moreover, the reaction of 4-aminoindole (**1m**) and **2q** gave the product **4mq** with 81% yield and 97% ee. 7-Aminoindole (**1n**) provided **4nq** in 22% yield and with 62% ee. While, simple aniline could not transform to the desired product **4oq** under the standard reaction conditions.

Then we focused on the investigation of simple anilines as the substrates (Fig. 3). Fortunately, *N*-alkyl anilines could be directly transformed to the desired products in one pot after a series of condition optimizations. For example, Product **4pq** was produced in 82% yield with 89% ee using (*S*)-**6b** as the catalyst, benzene as solvent, 5 Å molecular sieve as additive at 40 °C. Various linear alkyl groups, such as methyl, ethyl, *n*-butyl and allyl, substituted anilines could be transformed to the desired products with 91–92% ee. In addition, *N*-iso-propylaniline could give the product **4tq** in 89% yield with 84% ee. The multiple substituents could be tolerated, such as alkyl, halogen, electron donating and withdrawing groups. Methyl and methoxy at the *meta*-position of *N*-methylaniline could provide products **4wq** and **4xq** in excellent yields with 96% ee. *N*-Methylaniline bearing fluoro, chloro, bromo and iodo groups at the *meta*-position could deliver the desired products in excellent yields with good ee (**4xq-4Aq**, 89–93% ee). The *para* substituted *N*-methylaniline offered the product **4Cq** with 82% ee. The *ortho* position was linked to *N* atom with a six mumbled ring,

**Table 1 | Optimization of reaction conditions[a]**

| entry | CPA | solvent | yield (%)[b] | ee (%)[c] |
|---|---|---|---|---|
| 1 | (R)-5a | toluene | 95 | −45 |
| 2 | (R)-5b | toluene | 84 | −59 |
| 3 | (R)-5d | toluene | 90 | 28 |
| 4 | (R)-5e | toluene | 92 | 27 |
| 5[d] | (R)-5b | CH₂Cl₂ | 71 | −69 |
| 6[d] | (R)-5b | ODCB | 67 | −69 |
| 7[d] | (R)-5c | ODCB | 66 | −81 |
| 8[d] | (R)-5c | CH₂Cl₂ : ODCB = 4 : 1 | 70 | −88 |
| 9[defg] | (R)-5c | CH₂Cl₂ : ODCB = 4 : 1 | 59 | −91 |
| 10 | (R)-6a | toluene | 97 | −52 |
| 11 | (R)-6b | toluene | 95 | −68 |
| 12 | (R)-6c | toluene | 37 | −2 |
| 13[def] | (S)-6b | TBME | 99 | 87 |
| 14[def] | (S)-6b | AcO$^i$Pr | 99 | 88 |
| 15[defg] | (S)-6b | AcO$^i$Pr | 74 | 92 |

[a]Reaction conditions: (a) **1a** (0.05 mmol), **2a** (0.055 mmol), CPA (10 mol %), solvent (2 mL), rt, 40 h. **3aa** was isolated for the second step. (b) toluene (1 mL), 100 °C, 20 h. $^i$Pr = *iso*-propyl, ODCB = 1,2-dichlorobenzene, TBME = *tert*-butyl methyl ether. AcO$^i$Pr = *iso*-propyl acetate.
[b]Isolated yields.
[c]Determined by chiral HPLC analysis.
[d]Step b: TsOH·H₂O (1.0 equiv) was added in the second step at room temperature for 10 h.
[e]The **3aa** was not isolated and TsOH·H₂O was added directly in one-pot.
[f]7 d for step a.
[g]Solvent (4 mL).

products **4Fq-4Iq** could be obtained with 84–89% ee. An enantioenriched unnatural fluorescent amino acid (**4Jq**) was prepared starting from lysine. A variety of drug molecules (e.g., ibuprofen, isoxepac, indomethacin, and probenecid) were well tolerated, providing the desired 3-oxindoles in good yields with 90–91% ee (**4Kq-4Nq**).

**Mechanistic investigation**

To validate if the 3-hydroxyindolenine **3** was a key intermediate, the control experiment of **1a** and **2q** using (S)-**6b** as the catalyst was conducted (Fig. 4a). **3aq** was obtained in 99% yield with 98% ee. This result implied that this methodology could also be used for catalytic enantioselective synthesis of chiral 3-hydroxyindolenines, and the nucleophilic addition of *ortho*-C-H to carbonyl group might be the determining step of enantioselectivity. In addition, the isolated chiral **3aq** (98% ee) was transformed to the final product **4aq** with 97% ee after treated with TsOH·H₂O, which demonstrated that the exclusive 1,2-ester migration occurred after nucleophilic addition of *ortho*-C-H to

carbonyl group. With the diphenyl phosphate as the catalyst, only 47% yield of *rac*-**3aq** was obtained as well as ketimine **7** as the by-product. The isolated **7** could be transformed to the **4aq** with 98% ee under the standard conditions.

The condensations of a series of 2,3-diketoesters with 1-naphthylamine were performed to illustrate the impact of the electronic effect on the step of enantioselective *ortho* addition (Fig. 4b). A positive value (0.74) was observed based on the Hammett analysis, implying that the reaction rate of the 2,3-diketoester with electron-withdrawing groups at the *para* position to the central carbonyl was faster than that of substrates with electron-donating groups, indicating that the *ortho*-Friedel-Crafts addition to carbonyl group might be the rate-limiting step for the formation of 3-hydroxyindolenine **3aq**.

To elaborate the detailed reaction mechanism and control factor of the enantioselectivity, density functional theory (DFT) calculations[42,43] were conducted on the reaction 1-naphthylamine **1a**

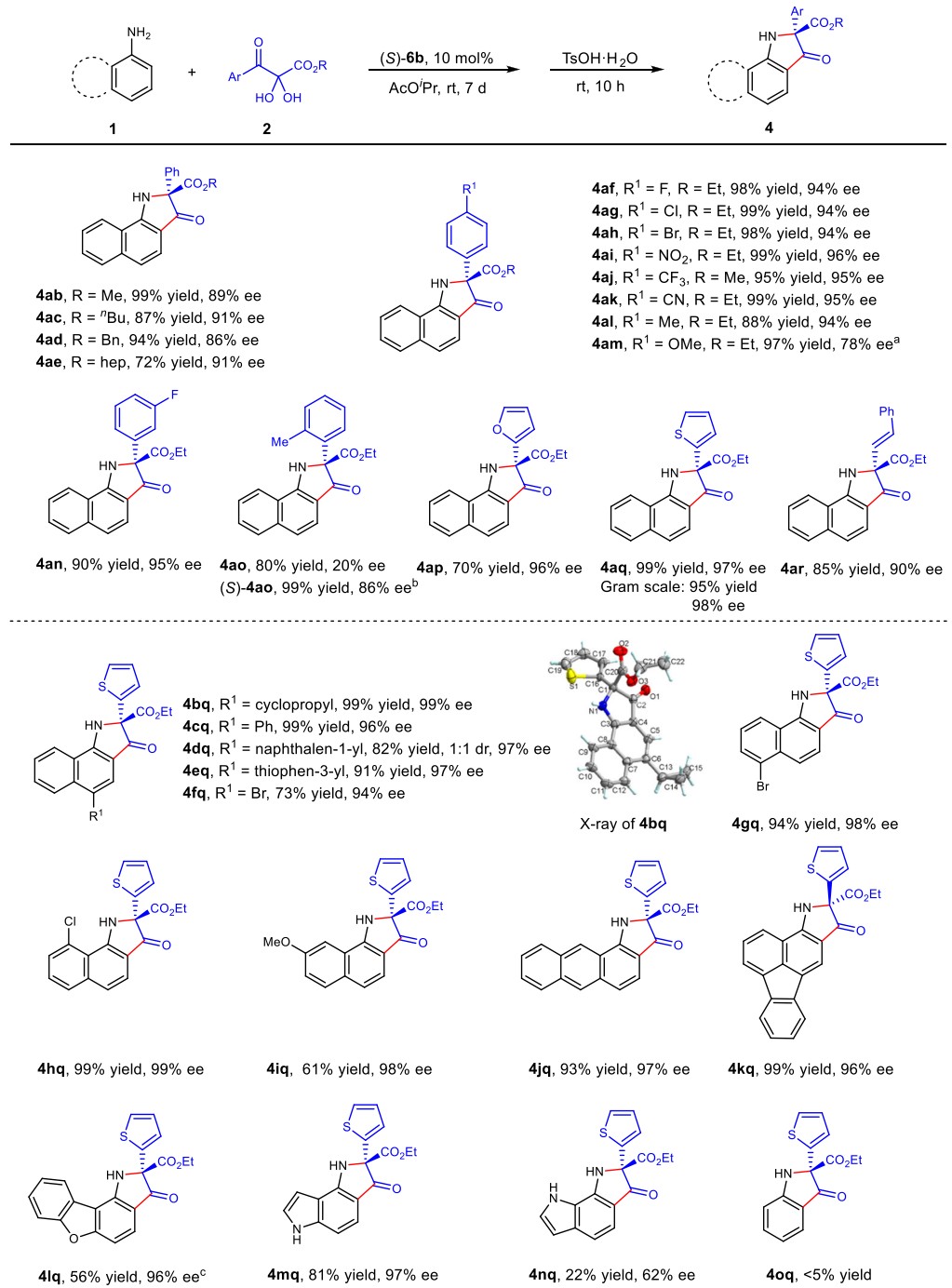

**Fig. 2 | Scope of primary arylamines and 2,3-diketoesters.** Reaction conditions: **1** (0.05 mmol), **2** (0.055 mmol), (*S*)-**6b** (10 mol %), AcO*i*Pr (4 mL), rt, 7 d, TsOH·H₂O (1.0 equiv) was added in one-pot at the second step at room temperature for 10 h. [a]AcO*i*Pr (0.5 mL). [b](*R*)-**5c** (10 mol %), CH₂Cl₂: ODCB (4 : 1, 0.5 mL), 40 °C. [c]AcO*i*Pr (1 mL).

and 2,3-diketoester **2q** in the presence of the CPA (*S*)-**6b** catalyst. The relative free energy profiles were calculated by M06-2X density functional, which was already parametrized to account for dispersion interaction[44]. As shown in Fig. 5, capture of 1-naphthylamine **1a** and 2,3-diketoester **2q** by CPA (*S*)-**6b** through hydrogen bonding successively could generate **Int5** with an endergonic free energy of 2.0 kcal/mol. The subsequent intramolecular nucleophilic addition of 1-naphthylamine **1a** at the carbonyl group of 2,3-diketoester **2q** via transition state **TS6-*RS*** furnished the zwitterionic intermediate **Int7-*RS***, requiring an activation-free energy of 9.3 kcal/mol. The dehydroaromatization then occurs rapidly via transition state **TS8-*R*** to generate the tertiary alcohol intermediate **Int9-*R***. Isomerization of **Int9-*R***

followed by intramolecular nucleophilic addition of amino at carbonyl group via transition state **TS11-*R*** with an activation energy barrier of 8.9 kcal/mol could afford ammonium **Int12-*R***. Deprotonation of **Int12-*R*** generates the diol **Int14-*R***. Eliminating one molecule of water of the resulting diol **Int14-*R*** gives the isolated 3-hydroxyindolenine **3aq**. Based on the calculated results, the nucleophilic addition of 1-naphthylamine **1a** at the carbonyl group of 2,3-diketoester **2q** via transition state **TS6-*RS*** could be considered as an irreversible and rate-determination step for the formation of 3-hydroxyindolenine **3aq**. Following that, we interrogated four possible enantioselective nucleophilic addition scenarios of 1-naphthylamine **1a** to the 2,3-diketoester **2q** via transition state **TS6-*RS***, **TS6-*RR***, **TS6-*SR***, and **TS6-*SS***

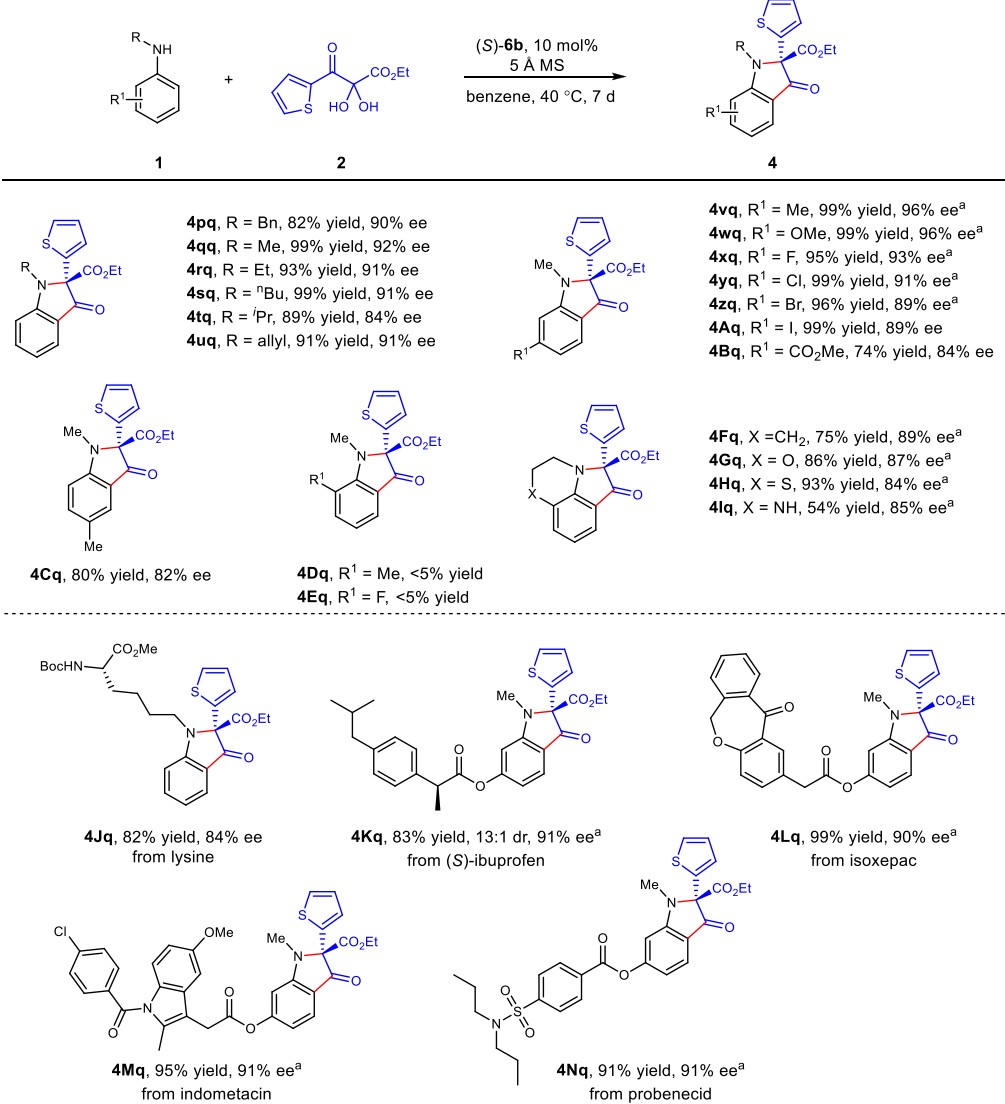

**Fig. 3 | Scope of anilines.** Reaction conditions: **1** (0.05 mmol), **2** (0.06 mmol), (S)-**6b** (10 mol %), 5 Å MS (50 mg), benzene (1 mL), 40 °C, 7 d. [a] 3 d.

(Fig. 6), which depicted that the nucleophilic addition of **1a** to **2q** via transition state **TS6-RS** was 4.1 kcal/mol lower than that of transition state **TS6-SR**, indicating that high enantioselectivity would be observed theoretically and experimentally.

To shine a light on the origin of the enantioselectivity, we further conducted the non-covalent Interaction (NCI) analysis for the key transition states **TS6-RS**, **TS6-RR**, **TS6-SR**, and **TS6-SS**. As shown in Fig. 6, the optimal matching factor for transition state **TS6-RS** not only originated from the π−π interaction (highlighted by blue circles) between the 9-anthryl groups in the arm of CPA catalyst and 1-naphthylamine **1a** and 2,3-diketoester **2q**, respectively, but also from the π−π interaction between naphthyl moiety of **1a** and thiazolyl moiety of **2a**. On the contrary, owing to hydrogen bonding interaction between CPA and 1-naphthylamine **1a** leading to an outward naphthyl moiety of **1a**, which erased the π−π interaction between the 9-anthryl groups of CPA catalyst and **1a**. Therefore, a relatively higher activation free energy would be assigned to the transition state **TS6-SR**. Similarly, the deficiency of π−π interaction between naphthyl moiety of **1a** and thiazolyl moiety of **2a** owing to the opposite direction of those two moieties in transition state **TS6-RR** and **TS6-SS**, resulting in unfavorable processes. Therefore, DFT calculation depicted that the π−π interaction plays a pivotal role in the enantioselectivity-determining process.

As shown in Fig. 7 DFT calculation was further employed to disclose the mechanism for the TsOH-catalyzed intramolecular 1,2-eater migration of 3-hydroxyindolenines **3aq**. The binding of TsOH with **3aq** through hydrogen bonding produced intermediate **Int15**. The following hydrogen transfer from TsOH to imine could generate zwitterionic intermediate **Int17**. Subsequently, the intramolecular 1,2-ester migration proceeds via transition state **TS18** to afford the product **4aq** accompanied by regenerating of TsOH with exergonic free energy of 9.5 kcal/mol. Optimized geometric for the transition states **TS18** showed that the bond length of the forming and breaking C-C bond was 2.01 and 2.01 Angstrom, respectively, suggesting that a concerted migration process could occur. Therefore, the chirality information of 3-hydroxyindolenine **3aq** could deliver completely to 3-oxindoles **4aq**, which was well consistent with experimental observations.

## Synthetic application

Recently, donor-acceptor energy transfer rigid system has been proved to be effective to enlarge the Stokes shifts[45,46]. The 2,2-disubstituted 3-oxindoles with amino and carbonyl groups as donor-acceptor system in a rigid ring might possess good photophysical properties. The fluorescence spectra of **4aq** and **4qq** in water were recorded (Fig. 8). The strong fluorescence was exhibited and the maximum emission wavelengths (493 and 534 nm) were recorded. The

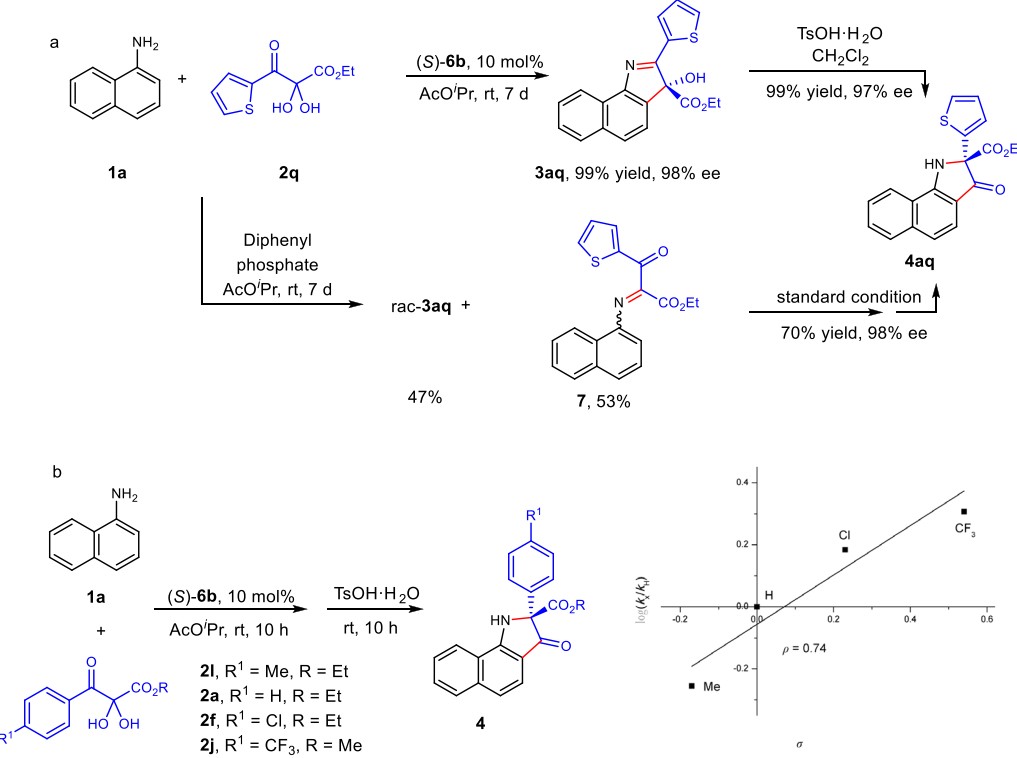

**Fig. 4 | Mechanism studies. a** Control experiments. **b** Hammett analysis of *ortho*-Friedel-Crafts addition of 1-naphthylamine to 2,3-diketoesters.

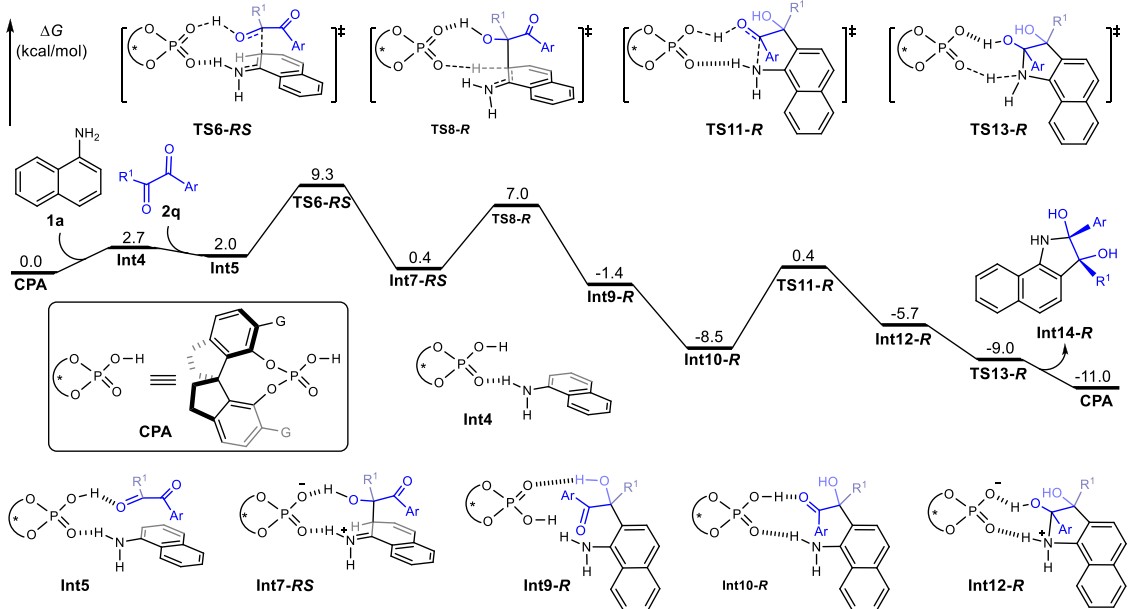

**Fig. 5 |** Calculated free energy profiles for the chiral phosphoric acid catalyzed *ortho*-Friedel-Crafts addition of 1-naphthylamine **1a** and 2,3-diketoester **2q** (G = 9-anthryl, R[1] = CO₂Et, Ar = thienyl). The values given in kcal/mol are the relative free energies calculated by the SMD (diethylether)/M06-2X/6-311+g(d,p)//SMD(diethylether)/M06-2X/6-31 g(d) method in diethylether.

stokes shifts were up to 75 and 105 nm respectively. Their photophysical properties in water would benefit their biological application.

Further transformations of enantioenriched 3-oxindoles were performed to illustrate the synthetic potential of this reaction (Fig. 9). Reduction of **4aq** with BH₃·DMS resulted in a chiral amino alcohol **8** with 90% ee. The following cyclization with 1,1′-carbonyldiimidazole (CDI) under DMAP catalysis afforded the polycyclic compound **9** with 92% ee. The catalytic hydrogenation of **4ar** delivered alkyl substituted

3-oxindole **10** with 94% ee. The reduced products **11** and **12** could be obtained respectively through controlling reaction time from **4qq**. Treated with Lawesson's reagent, indoline-3-thione **13** was afforded and exhibited a maximum absorption wavelength of 523 nm. All the resulted obtained above showed the advantage of the current protocol as it is difficultly accessible by other means.

In summary, we have developed an organocatalyzed de novo construction of chiral 3-hydroxyindolenines and 3-oxindoles with

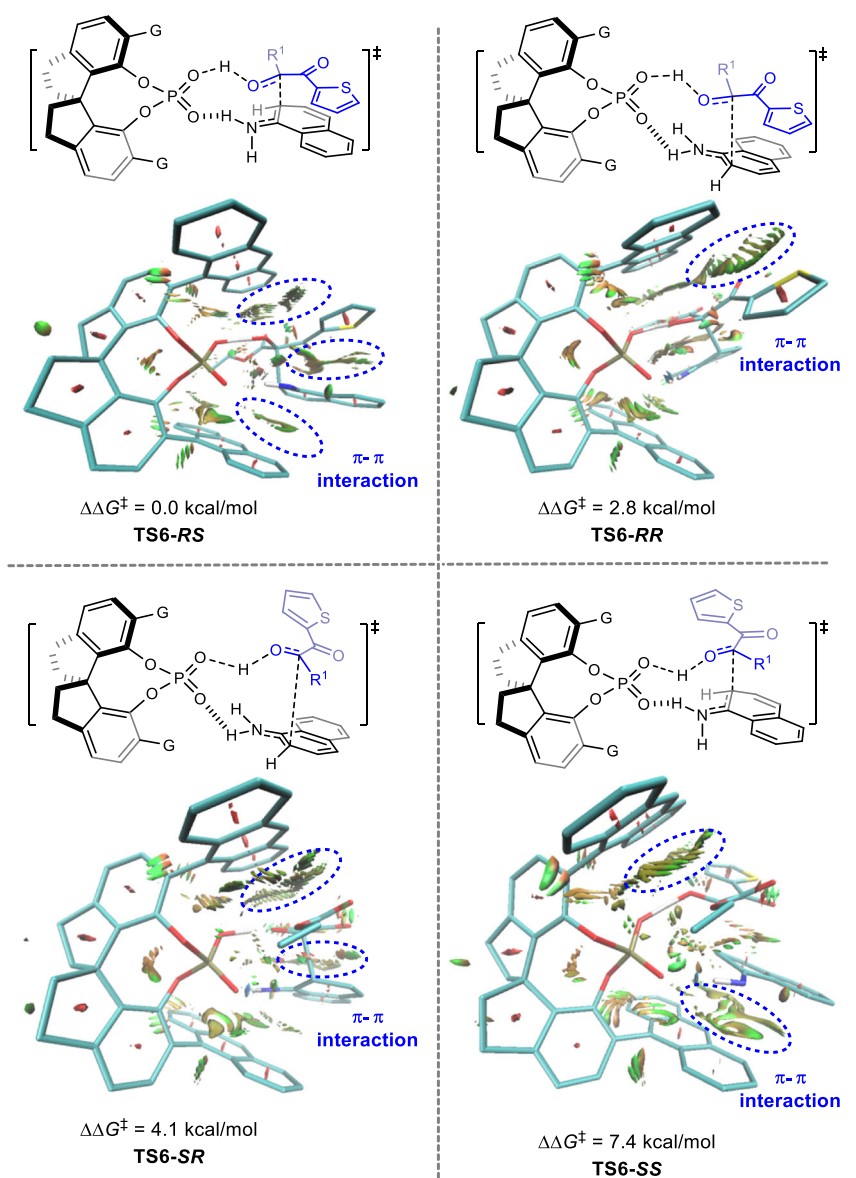

**Fig. 6 | Non-covalent Interaction (NCI) analysis for the key transition states TS6-*RS*, TS6-*RR*, TS6-*SR*, and TS6-*SS*.**

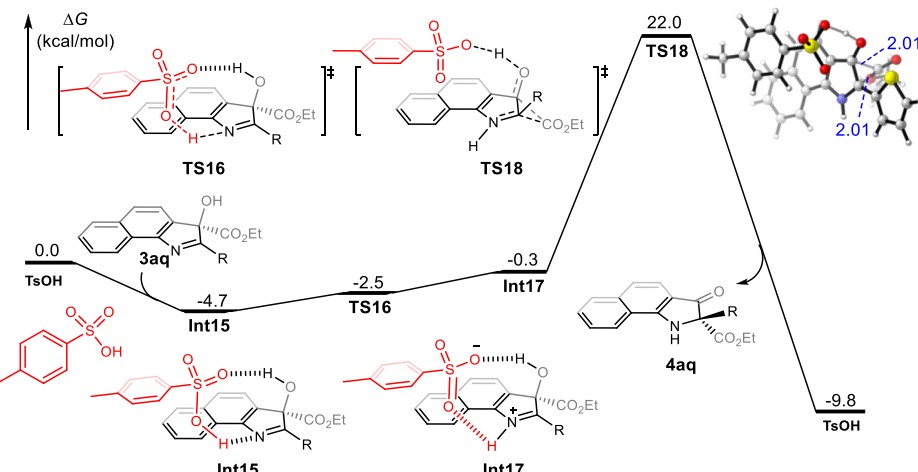

**Fig. 7 | Calculated free energy profiles for TsOH catalyzed intramolecular 1,2-ester migration of 3-hydroxyindolenines 3aq (R = thiazolyl).** The values given in kcal/mol are the relative free energies calculated by the SMD (diethylether)/ M06-2X/6-311+g(d,p)//SMD(diethylether)/M06-2X/6-31 g(d) method in diethylether. The blue values of geometric information are the bond lengths given in angstroms.

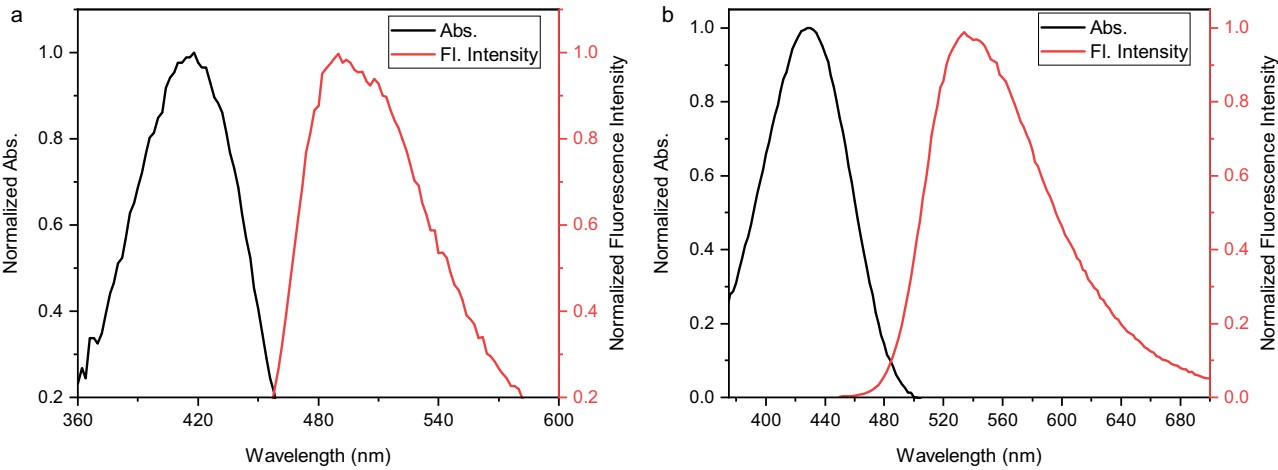

**Fig. 8 | Photophysical properties.** Absorption and fluorescence spectra of 4aq (**a**) and 4qq (**b**).

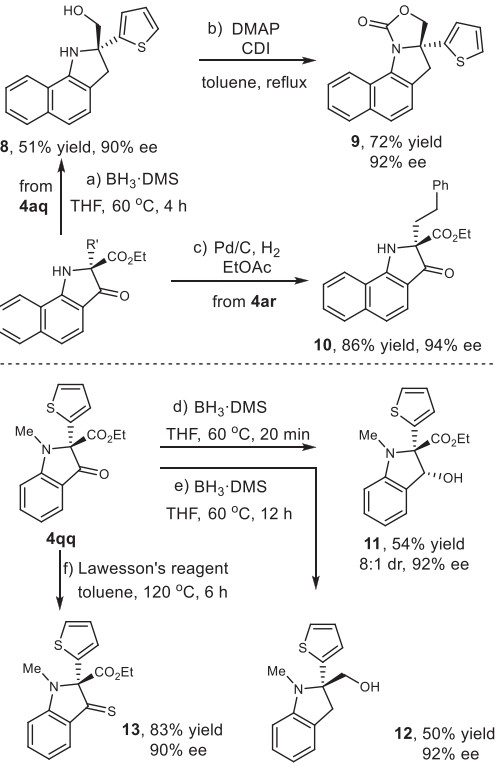

**Fig. 9 | Further chemical transformations. a** Reduction reaction to amino alcohol of **4aq**. DMS = dimethyl sulfide. **b** Cyclocarbonylation of **8**, DMAP = 4-dimethylaminopyridine, CDI = 1,1′-carbonyldiimidazole. **c** Alkene hydrogenation of **4ar**. **d** Carbonyl reduction of **4qq**. **e** Reduction reaction to amino alcohol of **4qq**. **f** Treated with Lawesson's reagent of **4qq**.

excellent yields and enantioselectivity starting from simple primary aryl amines and 2,3-diketoesters. The reaction was initiated by an intermolecular direct *ortho*-regioselective C-H addition to ketone carbonyl group. The following heteroanulation gave 3-hydroxyindolenines with a chiral tertiary carbon center. Followed by a rare enantioselective [1,2]-ester migration afforded the chiral 3-oxindoles. The DFT calculation depicted that the π − π interaction played a pivotal role in the enantioselectivity-determining process and a concerted 1,2-ester migration was demonstrated.

## Methods

### General procedure for de novo construction of chiral 3-oxindoles with primary amines

The amine **1** (0.05 mmol), 2,3-diketoester **2** (0.055 mmol) and (*S*)-**6b** (10 mol %) were dissolved in AcO$^i$Pr. The reaction mixture was stirred at room temperature for 7 d. Then the TsOH·H$_2$O (1.0 equiv) was added. The reaction mixture was stirred at room temperature for another 10 h. The solvent was removed in vacuo and the crude product was separated by flash column chromatography on silica gel to afford product **4**.

### General procedure for de novo construction of chiral 3-oxindoles with secondary amines

The amine **1** (0.05 mmol), 2,3-diketoester **2** (0.06 mmol), (*S*)-**6b** (10 mol %) and 5 Å MS (50 mg) was added in benzene. The reaction mixture was stirred at 40 °C for indicated time. Then the crude product was separated by flash column chromatography on silica gel to afford product **4**.

## Data availability

The authors declare that the data relating to the characterization of products, experimental protocols and the computational studies are available within this article and its Supplementary Information. Source data are provided in this paper. The X-ray crystallographic coordinates for structure **4bq** reported in this study have been deposited at the Cambridge Crystallographic Data Center (CCDC), under deposition numbers 2262643. These data can be obtained free of charge from The Cambridge Crystallographic Data Center via www.ccdc.cam.ac.uk/data_request/cif. Data can also be obtained from the corresponding author upon request. Source data are provided with this paper.

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

## Acknowledgements

We greatly acknowledge financial support from the Science and Technology department of Henan Province (232102310366 to Y.Wang, 242102310465 to H.C.) and Open Grant from the Pingyuan Laboratory (2023PY-OP-0208 to X.C.). Y. Wang. thanks Ph. D Shikun Jia (Zhengzhou University) and Prof. Hua Wu (Shanghai Jiao Tong University) for valuable discussions.

## Author contributions

Y.Wang and X.C. conceived the work. Y.Wang designed, conducted the experiments, analysed the data and wrote the manuscript. Y.Li conducted part of experiments under the supervision of Y.Wang. H.Chen performed the DFT calculations and wrote the calculation part of the manuscript. Y.Lan. and C.P. took part in the discussion. X.C and Y.Wu revised the manuscript.

## Competing interests

The authors declare no competing interests.
