## [Peer Review File · Nature Communications]

Enantioselective de novo construction of 3-oxindoles via organocatalyzed formal [3+2] annulation from simple arylaminesReviewers' Comments:

Reviewer #1:

Remarks to the Author:

This manuscript by Cui and co-workers described a protocol for the de novo construction of chiral 3-oxindoles bearing a quaternary chiral center using arylamines and diketoesters as starting materials. CPAs were used to promote the asymmetric ortho Friedel-Crafts alkylation of arylamine to give 3-hydroxyindolenines, which underwent [1,2]-ester migration to afford 3-oxindoles with excellent stereocontrol. Even though the reported chemistry disclosed a new method for the synthesis of these chiral indole derivatives, the research is too specialized for publication in Nat. Commun., and some issues need be addressed: 1) The scope of substrate in this study is too narrow. Only 1-naphthylamine and its analogues, such as 4-aminoindole, are applicable, 2-naphthylamine and aniline derivatives were not examined. In addition, only primary arylamines were used, what is the result of N-alkyl arylamines? Those results should be included in the manuscript. 2) The [1,2]-ester migration is interesting, however, the mechanism and driving force of this process was not studied. The absolute configuration of 3-hydroxyindolenine intermediate should also be given to elucidate the stereochemistry in this process. 3) In the SI, P95-96, the HPLC spectra indicated that there are two diastereoisomers for compound 4dq with a dr value of 1:1. Are there two stereogenic centers? This result should be discussed in the manuscript.

Reviewer #2:

Remarks to the Author:

In the manuscript, Cui and co-workers reported the organocatalyzed construction of chiral 3-oxindoles from primary 1-naphthyl-type amine and 2,3-diketoesters, through a sequential enantioselective ortho-FC reaction, condensation and 1,2-ester migration. A series of chiral 3-oxindoles bearing a tetrasubstituted stereogenic center was achieved in good yields and enantioselectivities. It is challenging to realize the intermolecular ortho-FC reaction of primary aryl amines and ketones, due to the side condensation reaction and para-selectivity.

Some interesting results were presented in the manuscript, and the direct construction of the related scaffolds, which may have potential applications, is challenging. This reviewer thinks that the current work might contain the novel findings meriting the publication on Nat. Commun., however, a number of issues might be addressed before decision is made, especially for the mechanism study.

1) In general, this manuscript was not well prepared. For example, the data in Table 1 were not clearly elucidated. According the formula in Table 1, two steps were required to give the final product 4aa, but the authors did not mentioned it in the text. In addition, TsOH was used for some entries, and no explanation was provided too. The authors should check the whole manuscript, guaranteeing it is written in a clearly understandable pattern.

2) The final enantioselective 1,2-ester migration process is intriguing; nevertheless, the suggested direct migration of ester group to electrophilic ketimine group might not be reasonable. Since TsOH was used for the ester migration reaction at rt in one-pot, did the chiral PA also play a role in this process? More control experiments might be conducted with the isolated chiral 3aq (Fig. 3): Will the ee value of chiral 3aq be changed when sole TsOH is used? Moreover, the authors are strongly encouraged to conduct some DFT calculations to illuminate the reaction pathway and the stereocontrol modes.

3) Will the transformation of imine product 7 be promoted by a chiral PA?

4) In general, the scope for both partners were relatively limited. How about 2-naphthylamine or even aniline-type ones? How about secondary amine substrates? Are the alkyl-substituted diketoesters applicable?

5) Significant ee losses were observed in the transformation of 4ag to 8. Why?

Reviewer #3:
Remarks to the Author:

Comments:

This manuscript by Cui and coworkers presents the use of a chiral phosphoric acid as catalyst for the enantioselective synthesis of chiral 3-oxindoles via ortho-Friedel-Crafts addition and novel [1,2]-ester migration. In particular, the [1,2]-ester migration reported in this paper is rarely seen in reactions. However, firstly, the chemistry presented in this communication is far from being very effective in terms of synthetic application. Meanwhile, the reaction time is too long. In terms of substrate, there are only polycyclic compounds, and there is no benzene as a substrate for reaction. On one hand, When the 2,3-diketoesters is a 2-substituted aromatic ring, the enantioselectivity is only 22% e.e.. On the other, the total number of substrates was insufficient to support publication in nature communication. Secondly, the title was chiral 3-oxindoles enantioselective synthesis. However, no direct synthesis of related compounds was found in this paper, basically all naphthalene indole compounds. Thirdly, there is insufficient evidence in mechanism research to demonstrate the formation process of chirality. Finally, the further transformation of the product is too simplistic. For all these reasons, I cannot recommend publication of this work in nature communication.

Other issues:

1) Why does the reaction catalyzed with CPA could selectively afford the desired product and the reaction catalyzed with diphenyl phosphate give the imine intermediate.

2) The reaction proceeds two-steps one pot to afford the final products. Why not stop in the first step to obtain the chiral imine product, which could further reduce to the chiral amine product? In this way, sufficient data could be obtained.

Other minor issues

The NMR spectrum should correspond to the serial number of each compound.

For compounds containing fluorine, their carbon spectrum splitting should be amplified.

The HPLC peak of compound 4jq is too low.

To the comments of Reviewer 1

This manuscript by Cui and co-workers described a protocol for the *de novo* construction of chiral 3-oxindoles bearing a quaternary chiral center using arylamines and diketoesters as starting materials. CPAs were used to promote the asymmetric ortho Friedel-Crafts alkylation of arylamine to give 3-hydroxyindolenines, which underwent [1,2]-ester migration to afford 3-oxindoles with excellent stereocontrol. Even though the reported chemistry disclosed a new method for the synthesis of these chiral indole derivatives, the research is too specialized for publication in *Nat. Commun.*, and some issues need be addressed:

Responses: The introduction of the manuscript was modified. The *de novo* construction of chiral 3-oxindoles would be interesting to interdisciplinary readership of *Nature Communications*.

1) The scope of substrate in this study is too narrow. Only 1-naphthylamine and its analogues, such as 4-aminoindole, are applicable, 2-naphthylamine and aniline derivatives were not examined. In addition, only primary arylamines were used, what is the result of *N*-alkyl arylamines? Those results should be included in the manuscript.

Responses: The *N*-alkyl arylamines as the substrates were examined. The results were added in the manuscript (page 8-9) and Table 3.

2) The [1,2]-ester migration is interesting, however, the mechanism and driving force of this process was not studied. The absolute configuration of 3-hydroxyindolenine intermediate should also be given to elucidate the stereochemistry in this process.

Responses: The absolute configuration of 3-hydroxyindolenine was identified in the Fig. 2. And the exclusive enantioselective [1,2]-ester migration was also demonstrated by the control experiments. “To validate if the 3-hydroxyindolenine **3** was a key intermediate, the control experiment of **1a** and **2q** using (S)-**6b** as the catalyst was conducted (Fig. 2a). **3aq** was obtained in 99% yield with 98% ee. This result implied that this methodology could also be used for catalytic enantioselective synthesis of chiral 3-hydroxyindolenines, and, the nucleophilic addition of *ortho*-C-H to carbonyl group might be the determining step of enantioselectivity. In addition, the isolated chiral **3aq** (98% ee) was transformed to the final product **4aq** with 97% ee after treated with TsOH·H₂O, which demonstrated that the exclusive 1,2-ester migration occurred after nucleophilic addition of *ortho*-C-H to carbonyl group. With the diphenyl phosphate as the catalyst, only 47% yield of **rac-3aq** was obtained as well as ketimine **7** as the by-product. The isolated **7** could be transformed to the **4aq** with 98% ee under the standard conditions.”

In addition, we conducted the DFT calculations (Fig. 5) for this process which supported the above results. “As shown in Fig. 5 DFT calculation was further employed to disclose the mechanism for the TsOH-catalyzed intramolecular 1,2-ester migration of 3-hydroxyindolenines **3aq**. The binding of TsOH with **3aq** through hydrogen bonding produced intermediate **Int10**. The following hydrogen transfer from TsOH to imine could generate zwitterionic intermediate **Int12**. Subsequently, the intramolecular 1,2-ester migration proceed *via* transition state **TS13** to afford the product **4aq** accompanied by regenerating of TsOH with an exergonic free energy of 9.5 kcal/mol. Optimized geometric for the transition states **TS13** showed that the bond length of the forming and breaking C-C bond was 2.01 and 2.01 angstrom, respectively, suggesting that a concerted migration process could occur. Therefore, the chirality information of 3-hydroxyindolenine **3aq** could deliver completely to 3-oxindoles **4aq**, which was well consistent with experimental observations.”

Fig. 5. Calculated free energy profiles for TsOH catalyzed intramolecular 1,2-ester migration of 3-hydroxyindolenines **3aq** (**R** = thiazolyl). The values given in kcal/mol are the relative free energies calculated by the SMD (diethylether)/M06-2X/6-311+g(d,p)//SMD(diethylether)/M06-2X/6-31g(d) method in diethylether. The blue values of geometric information are the bond lengths given in angstroms.

3) In the SI, P95-96, the HPLC spectra indicated that there are two diastereoisomers for compound **4dq** with a dr value of 1:1. Are there two stereogenic centers? This result should be discussed in the manuscript.

Responses: The compound **4dq** contain both a chiral quaternary carbon and an axially chiral center. The discussion was added into the manuscript. “Among them, product **4dq** with two stereogenic centers of a chiral quaternary carbon and an axially chiral center (1:1 dr)”

To the comments of Reviewer 2

In the manuscript, Cui and co-workers reported the organocatalyzed construction of chiral 3-oxindoles from primary 1-naphthyl-type amine and 2,3-diketoesters, through a sequential enantioselective ortho-FC reaction, condensation and 1,2-ester migration. A series of chiral 3-oxindoles bearing a tetrasubstituted stereogenic center was achieved in good yields and enantioselectivities. It is challenging to realize the intermolecular ortho-FC reaction of primary aryl amines and ketones, due to the side condensation reaction and para-selectivity.

Some interesting results were presented in the manuscript, and the direct construction of the related scaffolds, which may have potential applications, is challenging. This reviewer thinks that the current work might contain the novel findings meriting the publication on Nat. Commun., however, a number of issues might be addressed before decision is made, especially for the mechanism study.

Responses:

We thank the Reviewer 2 very much for the support and valuable comments!

We have addressed the issues including detailed mechanism study.

1) In general, this manuscript was not well prepared. For example, the data in Table 1 were not clearly elucidated. According the formula in Table 1, two steps were required to give the final product **4aa**, but the authors did not mention it in the text. In addition, TsOH was used for some entries, and no explanation was provided too. The authors should check the whole

manuscript, guaranteeing it is written in a clearly understandable pattern.

Responses: The whole manuscript, especially, introduction and investigation of reaction condition were revised. The two steps reaction introduction was added in page 4-5 and entries 5-9 in Table 1.

2) The final enantioselective 1,2-ester migration process is intriguing; nevertheless, the suggested direct migration of ester group to electrophilic ketimine group might not be reasonable. Since TsOH was used for the ester migration reaction at rt in one-pot, did the chiral PA also play a role in this process? More control experiments might be conducted with the isolated chiral **3aq** (Fig. 2): Will the ee value of chiral **3aq** be changed when sole TsOH is used? Moreover, the authors are strongly encouraged to conduct some DFT calculations to illuminate the reaction pathway and the stereocontrol modes.

Responses:

The exclusive enantioselective [1,2]-ester migration was demonstrated by the control experiments in Fig. 2a. The chiral PA could also catalyze the 1,2-ester migration, but it went very slowly under the room temperature. Hence, TsOH was add to accelerate this process. “To validate if the 3-hydroxyindolenine **3** was a key intermediate, the control experiment of **1a** and **2q** using (**S**)-**6b** as the catalyst was conducted (Fig. 2a). **3aq** was obtained in 99% yield with 98% ee. This result implied that this

methodology could also be used for catalytic enantioselective synthesis of chiral 3-hydroxyindolenines, and, the nucleophilic addition of *ortho*-C-H to carbonyl group might be the determining step of enantioselectivity. In addition, the isolated chiral **3aq** (98% ee) was transformed to the final product **4aq** with 97% ee after treated with TsOH·H₂O, which demonstrated that the exclusive 1,2-ester migration occurred after nucleophilic addition of *ortho*-C-H to carbonyl group. With the diphenyl phosphate as the catalyst, only 47% yield of **rac-3aq** was obtained as well as ketimine **7** as the by-product. The isolated **7** could be transformed to the **4aq** with 98% ee under the standard conditions.”

Fig. 2a Mechanism studies.

To elaborate the detailed reaction mechanism and controlling factor of the enantioselectivity, density functional theory (DFT) calculations were conducted on the reaction 1-naphthylamine **1a** and 2,3-diketoester **2q** in the presence of the CPA **(S)-6b** as the catalyst at the SMD (diethylether)/M06-2X/6-311+g(d,p)//SMD (diethylether)/M06-2X/6-

31g(d) level of theory. As shown in Fig. 3, capture of 1-naphthylamine **1a** and 2,3-diketoester **2q** by CPA (**S**)- **6b** through hydrogen bonding successively could generate **Int5** with an endergonic free energy of 2.0 kcal/mol. The subsequent intramolecular nucleophilic addition of 1-naphthylamine **1a** onto the central ketone carbonyl of 2,3-diketoester **2q** *via* transition state **TS6-*RS*** furnished the zwitterionic intermediate **Int7-*RS***, requiring an activation free energy of 9.3 kcal/mol. The dehydro-aromatization then could occur rapidly *via* transition state **TS8-*R*** to generate the tertiary alcohol intermediate **Int9-*R***. The calculated results depicted that the nucleophilic addition of 1-naphthylamine **1a** onto the central ketone carbonyl of 2,3-diketoester **2q** could be considered as the enantioselectivity and rate determination step. Following that, we interrogated four possible enantioselective nucleophilic addition scenarios of 1-naphthylamine **1a** to the 2,3-diketoester **2q** *via* transition state **TS6-*RS***, **TS6-*RR***, **TS6-*SR***, and **TS6-*SS*** (Fig. 4), which depicted that the nucleophilic addition of **1a** to **2q** *via* transition state **TS6-*RS*** was 4.1 kcal/mol lower than that of transition state **TS6-*SR***, indicating that high enantioselectivity would be observed theoretically and experimentally.

Fig. 3. Calculated free energy profiles for the chiral phosphoric acid catalyzed *ortho*-Friedel-Crafts addition of 1-naphthylamine **1a** and 2,3-diketoester **2q** ($G = 9\text{-anthryl}$, $R^1 = \text{CO}_2\text{Et}$). The values given in kcal/mol are the relative free energies calculated by the SMD (diethylether)/M06-2X/6-311+g(d,p)//SMD(diethylether)/M06-2X/6-31g(d) method in diethylether.

To shine a light on the origin of the enantioselectivity, we further conducted the non-covalent Interaction (NCI) analysis for the key transition states **TS6-RS**, **TS6-RR**, **TS6-SR**, and **TS6-SS**. As shown in Fig. 4, the optimal matching factor for transition state **TS6-RS** not only originated from the $\pi-\pi$ interaction (highlighted by blue circles) between the 9-anthryl groups in the arm of CPA catalyst and 1-naphthylamine **1a** and 2,3-diketoester **2q**, respectively, but also from the $\pi-\pi$ interaction between naphthyl moiety of **1a** and thiazolyl moiety of **2a**. On the contrary, owing to hydrogen bonding interaction between CPA and 1-naphthylamine **1a** leading to an outward naphthyl moiety of **1a**, which erased the $\pi-\pi$ interaction between the 9-

anthryl groups of CPA catalyst and **1a**. Therefore, a relatively higher activation free energy would be assigned to the transition state **TS6-SR**. Similarly, the deficiency of π - π interaction between naphthyl moiety of **1a** and thiazolyl moiety of **2a** owing to the opposite direction of those two moieties in transition state **TS6-RR** and **TS6-SS**, resulting in unfavorable processes. Therefore, DFT calculation depicted that the π - π interaction plays a pivotal role in the enantioselectivity-determining process.

Fig. 4. Non-covalent Interaction (NCI) analysis for the key transition

states TS6-RS, TS6-RR, TS6-SR, and TS6-SS.

In addition, we conducted the DFT calculations (Fig. 5) for this process which supported the above results. “As shown in Fig. 5 DFT calculation was further employed to disclose the mechanism for the TsOH-catalyzed intramolecular 1,2-ester migration of 3-hydroxyindolenines **3aq**. The binding of TsOH with **3aq** through hydrogen bonding produced intermediate **Int10**. The following hydrogen transfer from TsOH to imine could generate zwitterionic intermediate **Int12**. Subsequently, the intramolecular 1,2-ester migration proceed *via* transition state **TS13** to afford the product **4aq** accompanied by regenerating of TsOH with an exergonic free energy of 9.5 kcal/mol. Optimized geometric for the transition states **TS13** showed that the bond length of the forming and breaking C-C bond was 2.01 and 2.01 angstrom, respectively, suggesting that a concerted migration process could occur. Therefore, the chirality information of 3-hydroxyindolenine **3aq** could deliver completely to 3-oxindoles **4aq**, which was well consistent with experimental observations.”

Fig. 5. Calculated free energy profiles for TsOH catalyzed intramolecular 1,2-ester migration of 3-hydroxyindolenines **3aq** ($R = \text{thiazolyl}$). The values given in kcal/mol are the relative free energies calculated by the SMD (diethylether)/M06-2X/6-311+g(d,p)//SMD(diethylether)/M06-2X/6-31g(d) method in diethylether. The blue values of geometric information are the bond lengths given in angstroms.

3) Will the transformation of imine product **7** be promoted by a chiral PA?

Responses: The isolated imine **7** could also be transformed to the **4aq** with 98% ee under the standard conditions (Fig. 2).

4) In general, the scope for both partners were relatively limited. How about 2-naphthylamine or even aniline-type ones? How about secondary amine substrates? Are the alkyl-substituted diketoesters applicable?

Responses: *N*-alkyl anilines as the substrates of were also examined and added in the manuscript (page 8-9) and Table 3.

The alkenyl substituted diketoesters was applicable and **4ar** was obtained in 85% yield with 90% ee (Table 2). This product **4ar** could be transformed to the alkyl substituted 3-oxindole **10** with 94% ee (Fig. 7).

5) Significant ee losses were observed in the transformation of **4ag** to **8**. Why?

Responses: We conducted the same transformation for *N*-alkyl-substituted 3-oxindole **4qq** (Fig. 7). The ee losses was not observed. Thus, the ee losses might be relevant with the unknown racemization in the reduction of the *N*-non-protected chiral 3-oxindole.

To the comments of Reviewer 3

This manuscript by Cui and coworkers presents the use of a chiral phosphoric acid as catalyst for the enantioselective synthesis of chiral 3-oxindoles via ortho-Friedel-Crafts addition and novel [1,2]-ester migration. In particular, the [1,2]-ester migration reported in this paper is rarely seen in reactions. However, firstly, the chemistry presented in this communication is far from being very effective in terms of synthetic application. Meanwhile, the reaction time is too long. In terms of substrate, there are only polycyclic compounds, and there is no benzene as a substrate for reaction. On one hand, When the 2,3-diketoesters is a 2-substituted aromatic ring, the enantioselectivity is only 22% e.e. On the other, the total number of substrates was insufficient to support publication in nature communication.

Responses: Gratifyingly, when using (**R**)-**5c** as the catalyst, 4:1 ratio of CH_2Cl_2 and ODCB as the solvent, (**S**)-**4ao** was obtained in excellent yield

with 86% ee. *N*-Alkyl anilines were also examined as shown in Table 3 (page 8-9). Thus, more than 55 substrates could be applied to this protocol.

Secondly, the title was chiral 3-oxindoles enantioselective synthesis. However, no direct synthesis of related compounds was found in this paper, basically all naphthalene indole compounds.

Responses: The direct synthesis of chiral 3-oxindoles were shown in Table 3 with anilines substrates.

Thirdly, there is insufficient evidence in mechanism research to demonstrate the formation process of chirality. Finally, the further transformation of the product is too simplistic. For all these reasons, I cannot recommend publication of this work in nature communication.

Responses: The detailed mechanistic investigation and DFT calculations have been conducted. And the results were added in pages 9-15. The more transformations were investigated and the results were added pages 15-17.

Other issues:

1) Why does the reaction catalyzed with CPA could selectively afford the desired product and the reaction catalyzed with diphenyl phosphate give

the imine intermediate.

Responses: Actually, the reversible imine formation was both exist in the reactions with CPA or diphenyl phosphate as catalyst. The isolated imine **7** could also be transformed to the **4aq** with 98% ee under the standard conditions (Fig. 2). The imine was isolated with diphenyl phosphate as catalyst because of the slower reaction rate and insufficient conversion.

2) The reaction proceeds two-steps one pot to afford the final products. Why not stop in the first step to obtain the chiral imine product, which could further reduce to the chiral amine product? In this way, sufficient data could be obtained.

Responses: The first step product **3aq** could also be isolated and identified.

The reduction of **3aq** resulted in indole compound.

Other minor issues

The NMR spectrum should correspond to the serial number of each compound.

Responses: The compounds numbers have been located at the bottom of the spectrum. For instance:

For compounds containing fluorine, their carbon spectrum splitting should be amplified.

Responses: For the compounds containing fluorine (**2f**, **2j**, **2n**, **4af**, **4aj**, **4an** and **4xq**), the carbon spectrum splitting were amplified:

We deeply appreciate your reconsideration of our manuscript, and we are looking forward to receiving the comments from the reviewers and yourself. If you have any queries, please don't hesitate to contact us.

Sincerely yours,

Xiuling Cui

Reviewers' Comments:

Reviewer #2:

Remarks to the Author:

In the revised manuscript, the authors have significantly expanded the substrate scope; in particular, the successful application of N-alkyl anilines to construct the expected products is impressive.

Moreover, DFT calculations were conducted to elucidate the whole reaction process, and an unusual 1,2-ester migration is interesting. Considering the progress has been made, this reviewer thinks the current work merits the publication on Nat. Commun.

There are many spelling and grammar errors in the text. The authors should carefully polish the manuscript accordingly. For example, in the abstract section, "unprecedented".

Reviewer #3:

Remarks to the Author:

The authors have addressed all the concerns such as the substrate scopes and the reaction mechanism. Meanwhile, the quality of the manuscript have been improved based on the suggestions of the referees. Therefore, I think it's suitable for nature communications.

Reviewer #4:

Remarks to the Author:

During the revision of the manuscript, the authors have performed extensive DFT calculations to propose a reaction mechanism and to identify the origin of the enantioselectivity. Certainly, these additional studies would strength the work, but I am not convinced that they would change the overall interest for a broad audience of Nature Communications. This is a specific reaction mechanism that is not related to other organic reactions, nor the explanation of the enantioselectivity from pi-pi interactions, are novel. Besides these considerations, in my opinion, there are also some relevant issues on the discussion of the DFT results that must be considered during revision of the manuscript.

Figure 4 compares different enantiomeric pathways, identifying pi-pi interactions as key factors governing enantioselectivity. These types of non-bonding interactions are not well reproduced by classical DFT functionals, and classical dispersion corrections are normally included in the calculations. Here, the authors use M06-2X functional without explicit dispersion. Thus, the authors should say a word on the quantitative performance of M06-2X functional for evaluating pi-pi interactions, because they are crucial for understanding enantioselectivity.

The assessment of enantioselectivity is based on the relative free-energies of isomeric transition states TS6. This assumes Curtin-Hammett conditions: rapid pre-equilibria of reactants and irreversible reaction. The energy values of the energy profile in Figure 3 do not indicate an irreversible reaction, because the low energy barriers and similar relative energies of the intermediates. It is true that the reaction could become irreversible from Int9. However, the part of the mechanism connecting the so-called Int9 and Int10 is not characterized. Therefore, the reversibility or irreversibility of the proposed enantioselectivity-determining step cannot be concluded. To sum up, two aspects should be further discussed: a) the irreversibility of the proposed enantioselectivity-determining step, and b) the description of the full mechanism connecting the reactants with the products.

In line 147, the authors state: "to clarify the enantioselectivity ortho addition as the turnover-limiting of this reaction". However, according to current DFT data, the rate-determining process corresponds to the energy difference between TS13 and Int10.

Thank you very much for your positive decision on our manuscript titled **“Enantioselective *de novo* construction of 3-oxindoles via organocatalyzed formal [3+2] annulation from simple arylamines”**. According the comments from reviewers, this manuscript was revised. The responses to the comments and the corresponding changes are disclosed as follows:

To the comments of Reviewer 2

In the revised manuscript, the authors have significantly expanded the substrate scope; in particular, the successful application of N-alkyl anilines to construct the expected products is impressive. Moreover, DFT calculations were conducted to elucidate the whole reaction process, and an unusual 1,2-ester migration is interesting. Considering the progress has been made, this reviewer thinks the current work merits the publication on Nat. Commun.

There are many spelling and grammar errors in the text. The authors should carefully polish the manuscript accordingly. For example, in the abstract section, “unprecedented”.

Responses: We sincerely appreciate for your kind comments. The manuscript was carefully polished. For instance, in the abstract, “unprecedented” was changed to “unprecedented”. In the last line, page 11,

“could occur” was changed to “occurs”. In page 15, “wavelength” was changed to “wavelengths” and “stokes shift” was changed to “stokes shifts”.

To the comments of Reviewer 3

The authors have addressed all the concerns such as the substrate scopes and the reaction mechanism. Meanwhile, the quality of the manuscript have been improved based on the suggestions of the referees. Therefore, i think it's suitable for nature communications.

Responses: Thanks for your kind comments.

To the comments of Reviewer 4

During the revision of the manuscript, the authors have performed extensive DFT calculations to propose a reaction mechanism and to identify the origin of the enantioselectivity. Certainly, these additional studies would strength the work, but I am not convinced that they would change the overall interest for a broad audience of Nature Communications. This is a specific reaction mechanism that is not related to other organic reactions, nor the explanation of the enantioselectivity from pi-pi interactions, are novel. Besides these considerations, in my opinion, there are also some relevant issues on the discussion of the DFT results that must be considered during revision of the manuscript.

Responses: Thanks for your favorable comments on our manuscript. We would like to emphasize that the synthesis of 3-oxindoles bearing chiral at the C2 position are of paramount importance in chemical community, since they are ubiquitous structural motifs that are found in natural products and biologically active compounds and also play a significant role in a variety of synthetic transformations. However, the existing reaction modes in literatures mainly rely on multistep reactions starting from 2-substituted indoles (Ref 10-26), the *de novo* construction of chiral 3-oxindoles from easily available starting materials as well as simple operation has been highly desired. In this work, the ready availability of arylamines were employed as the starting materials with 2,3-diketoesters via ortho-C-H addition to directly construct 3-hydroxyindolenines with chiral tertiary alcohols. This protocol offers a flexible platform to access a wide array of chiral 3-oxindoles with good yields, excellent functional group tolerance, and enantioselectivity. The rapid assembly of biologically valuable chiral 3-oxindole derivatives (**4Jq-4Nq**) confirmed the compatibility and practicability of this methodology. Moreover, the π - π interaction between substrates and catalyst, which leads to the observed enantioselectivity, was revealed by detailed mechanistic experiments and DFT calculations, which might provide an avenue not only for the construction of diversity and complexity chiral oxindole derivatives but also for CPA-catalyzed enantioselectivity transformations. In addition, incorporating the

reviewer's constructive comments, we have revised the manuscript and the detailed responses are listed as follows:

1) Figure 4 compares different enantiomeric pathways, identifying pi-pi interactions as key factors governing enantioselectivity. These types of non-bonding interactions are not well reproduced by classical DFT functionals, and classical dispersion corrections are normally included in the calculations. Here, the authors use M06-2X functional without explicit dispersion. Thus, the authors should say a word on the quantitative performance of M06-2X functional for evaluating pi-pi interactions, because they are crucial for understanding enantioselectivity.

Responses: According to your suggestion, the sentence "The relative free energy profiles were calculated by M06-2X density functional, which was already parametrized to account for dispersion interaction." was added in page 11.

2) The assessment of enantioselectivity is based on the relative free-energies of isomeric transition states TS6. This assumes Curtin-Hammett conditions: rapid pre-equilibria of reactants and irreversible reaction. The energy values of the energy profile in Figure 3 do not indicate an irreversible reaction, because the low energy barriers and similar relative energies of the intermediates. It is true that the reaction could become irreversible from Int9. However, the part of the mechanism connecting the so-called Int9 and Int10 is not characterized. Therefore, the reversibility or

irreversibility of the proposed enantioselectivity-determining step cannot be concluded. To sum up, two aspects should be further discussed: a) the irreversibility of the proposed enantioselectivity-determining step, and b) the description of the full mechanism connecting the reactants with the products.

Responses: Thanks for your valuable suggestion, which helps us improve the quality of the manuscript. To clarify that the intramolecular nucleophilic C-H addition of 1-naphthylamine **1a** at the carbonyl group of 2,3-diketoester *via* transition state **TS6-RS** is an irreversible step, the full mechanism was further taken into consideration. The calculated results are as follows and also the figure 3 was revised. The calculated full energy profiles depicted that the activation free energy of nucleophilic addition *via* **TS6-RS** is 9.3 kcal/mol, which is higher than that of the subsequent transformations of its generated zwitterionic intermediate **Int7-RS**. Therefore, the nucleophilic addition step could be considered an irreversible and rate-determination step for the formation of 3-hydroxyindolenine **3aq**.

Fig. 3.

The related description was reflected in page 12: “Isomerization of **Int9-R** followed by intramolecular nucleophilic addition of amino at carbonyl group *via* transition state **TS11-R** with an activation energy barrier of 8.9 kcal/mol could afford ammonium **Int12**. Deprotonation of **Int12** generates the diol **Int14-R**. Eliminating one molecule of water from the resulting diol **Int14-R** gives the isolated 3-hydroxyindolenine **3aq**. Based on the calculated results, the nucleophilic addition of 1-naphthylamine **1a** at the carbonyl group of 2,3-diketoester **2q** via transition state **TS6-RS** could be considered as an irreversible and rate-determination step for the formation of 3-hydroxyindolenine **3aq**.”

3) In line 147, the authors state: “to clarify the enantioselectivity ortho addition as the turnover-limiting of this reaction”. However, according to current DFT data, the rate-determining process corresponds to the energy

difference between TS13 and Int10.

Responses: This conclusion “ortho-Friedel-Crafts addition to carbonyl group might be the rate-limiting step for the formation of 3-hydroxyindolenine **3aq**” was drawn from the Hammett analysis (Figure 2b). Based on this fact, the corresponding description was revised as follows:

“The condensations of a series of 2,3-diketoesters with 1-naphthylamine were performed to illustrate the impact of the electronic effect on the step of enantioselective ortho addition (Fig. 2b). A positive value (0.74) was observed based on the Hammett analysis, implying that the reaction rate of the 2,3-diketoester with electron-withdrawing groups at the *para* position to the central carbonyl was faster than that of electron-donating groups, indicating that the ortho-Friedel-Crafts addition to carbonyl group might be the rate-limiting step for the formation of 3-hydroxyindolenine **3aq**.”

The changes were highlighted by yellow color in revised manuscript and supporting information. Please contact with us freely if there is other information to be addressed.

Reviewers' Comments:

Reviewer #4:

Remarks to the Author:

The revision of the manuscript has clarified my main concerns related with DFT calculations. The new energy profile reported in Figure 3 indicates that transition state TS6 corresponds to an irreversible step, and corresponds to the enantioselectivity determining step under Curti-Hammet conditions. The new text clarifies that the discussion of rate-determining process refers to the formation of product 3aq. My only suggestion here refers to Figure 2b, in which the authors could note more clearly that the Hammet plot corresponds to the first of the two consecutive reactions reported. Still, I see that the main interest of the manuscript does not relay in DFT mechanistic characterization, but in the synthetic utility of the reaction that has been positively evaluated by more competent reviewers. Thus, I am glad to support the publication of this work in Nat. Commun. Journal.

Thank you very much for your acceptance on our manuscript titled “**Enantioselective *de novo* construction of 3-oxindoles via organocatalyzed formal [3+2] annulation from simple arylamines**”. According the comments from reviewer 4, this manuscript was revised. The responses to the comments and the corresponding changes are disclosed as follows:

To the comments of Reviewer 4

Reviewer #4 (Remarks to the Author):

Comments: The revision of the manuscript has clarified my main concerns related with DFT calculations. The new energy profile reported in Figure 3 indicates that transition state TS6 corresponds to an irreversible step, and corresponds to the enantioselectivity determining step under Curti-Hammet conditions. The new text clarifies that the discussion of rate-determining process refers to the formation of product 3aq. My only suggestion here refers to Figure 2b, in which the authors could note more clearly that the Hammet plot corresponds to the first of the two consecutive reactions reported. Still, I see that the main interest of the manuscript does not relay in DFT mechanistic characterization, but in the synthetic utility of the reaction that has been positively evaluated by more competent

reviewers. Thus, I glad to support the publication of this work in Nat. Commun. Journal.

Responses: We sincerely thank this reviewer for his/her positive comments on our work. To make it clearer that the Hammett plot corresponds to the first of two consecutive reactions reported, the caption of Figure 2b (Figure 4b in revised manuscript) was revised as “Hammett analysis of the *ortho*-Friedel-Crafts addition of 1-naphthylamine to 2,3-diketoesters”.